# The Therapeutic Potential of Naturally Occurring Peptides in Counteracting SH-SY5Y Cells Injury

**DOI:** 10.3390/ijms231911778

**Published:** 2022-10-04

**Authors:** Renata Perlikowska, Joana Silva, Celso Alves, Patrícia Susano, Rui Pedrosa

**Affiliations:** 1Department of Biomolecular Chemistry, Faculty of Medicine, Medical University of Lodz, 92-215 Lodz, Poland; 2MARE—Marine and Environmental Sciences Centre, Politécnico de Leiria, 2520-630 Peniche, Portugal; 3MARE—Marine and Environmental Sciences Centre, ESTM, Politécnico de Leiria, 2520-641 Peniche, Portugal

**Keywords:** opioid peptides, 6-hydroxydopamine, neuronal survival, neuroprotection, antioxidant activity, apoptosis

## Abstract

Peptides have revealed a large range of biological activities with high selectivity and efficiency for the development of new drugs, including neuroprotective agents. Therefore, this work investigates the neuroprotective properties of naturally occurring peptides, endomorphin-1 (EM-1), endomorphin-2 (EM-2), rubiscolin-5 (R-5), and rubiscolin-6 (R-6). We aimed at answering the question of whether well-known opioid peptides can counteract cell injury in a common in vitro model of Parkinson’s disease (PD). Antioxidant activity of these four peptides was evaluated by the 2-diphenyl-1-picrylhydrazyl radical (DPPH) scavenging activity, oxygen radical absorbance capacity (ORAC), and ferric-reducing antioxidant power (FRAP) assays, while neuroprotective effects were assessed in a neurotoxic model induced by 6-hydroxydopamine (6-OHDA) in a human neuroblastoma cell line (SH-SY5Y). The mechanisms associated with neuroprotection were investigated by the determination of mitochondrial membrane potential (MMP), reactive oxygen species (ROS) production, and Caspase-3 activity. Among the tested peptides, endomorphins significantly prevented neuronal death induced by 6-OHDA treatment, decreasing MMP (EM-1) or Caspase-3 activity (EM-2). Meanwhile, R-6 showed antioxidant potential by FRAP assay and exhibited the highest capacity to recover the neurotoxicity induced by 6-OHDA via attenuation of ROS levels and mitochondrial dysfunction. Generally, we hypothesize that peptides’ ability to suppress the toxic effect induced by 6-OHDA may be mediated by different cellular mechanisms. The protective effect caused by endomorphins results in an antiapoptotic effect (mitochondrial protection and decrease in Caspase-3 activity), while R-6 potency to increase a cell’s viability seems to be mediated by reducing oxidative stress. Our results may provide new insight into neurodegeneration and support the short peptides as a potent drug candidate to treat PD. However, further studies should be conducted on the detailed mechanisms of how tested peptides could suppress neuronal injuries.

## 1. Introduction

The term neuroprotection applies to all possible approaches to secure neuronal structure and/or function, limit nerve death, and protect the central nervous system from premature degeneration and other causes of nerve cell death. In recent years, when neurodegenerating disorders have been the leading causes of severe long-term disability, research into neuroprotective therapies has rapidly progressed. Increasing interest has recently focused on determining whether several well-known natural or synthetic compounds may exert neuroprotective actions in the developing, adult, and ageing nervous system. In this area of research, natural and modified peptides appear as promising candidates. Peptides are substances with multidirectional biological activity, with high selectivity and efficiency [1,2]. At the same time, they are relatively safe and well tolerated and could be considered not only as a drug, but also as functional food or nutraceuticals. Generally, well-known therapeutic peptides may act as hormones, growth factors, neurotransmitters, ion channel ligands, or anti-infective agents. Their mechanism of action is associated with binding to cell surface receptors and triggering intracellular effects, such as antioxidant, anti-inflammatory, antihypertensive, antithrombotic, antiadipogenic, anticancer, and immunomodulatory [2]. 

Here, the author will deal with the evidence proving that four naturally occurring opioid peptides may have neuroprotective activity in the 6-hydroxydopamine (6-OHDA)-injury model that has been exploited as an experimental model to study Parkinson’s disease (PD) [3]. Due to its similarity in molecular structure to dopamine, 6-OHDA shows a high affinity for the dopamine transporter. Therefore, it selectively destroys dopaminergic/catecholaminergic neurons. Once inside the neuron, 6-OHDA accumulates and undergoes nonenzymatic auto-oxidation, promoting reactive oxygen species (ROS) formation, and leading to mitochondrial dysfunction [4]. Consequently, loss of the mitochondrial transmembrane potential can result in the rupture of the outer mitochondrial membrane and the release of proapoptotic proteins from the nucleus, leading to cell death through activation of the intrinsic apoptosis pathway. Thus, cell death induced by the 6-OHDA is also associated with activation of Caspase-3, Bax/Bcl-2 ratio, cytochrome c, and DNA fragmentation. According to evidence, the human neuroblastoma cell line (SH-SY5Y) selected in this study shares some morphological, neurochemical, and electrophysiological properties with normal neurons and is characterized by its low differentiation, pyramidal shape, and obvious axon. The SH-SY5Y cells have been extensively used as a model to study the neurotoxicity of numerous stimulants in vitro [5,6].

Herein, the in vitro neuroprotective activity of endogenous opioid peptides, endomorphin-1 (EM-1, Tyr-Pro-Trp-Phe-NH_2_) and endomorphin-2 (EM-2, Tyr-Pro-Phe-Phe-NH_2_), and plant-derived peptides, rubiscolin-5 (R-5, Tyr-Pro-Leu-Asp-Leu) and rubiscolin-6 (R-6, Tyr-Pro-Leu-Asp-Leu-Phe) (Figure 1) was characterized. Endomorphins, identified in the bovine brain and human cortex, contain the Tyr-Pro-Phe/Trp sequence at the N-terminus, while C-terminus is amidated [7]. Till now, they have been shown to exist in the rodent, primate, and human central nervous system and peripheral nervous system. Both opioids bind preferentially to µ-opioid receptors, thereby activating G-proteins, resulting in regulation of gastrointestinal motility, controlling acute and chronic pain, including neuropathic pain, cancer pain, and inflammatory pain. Endomorphins effect on the vascular systems, and neuroendocrine, and limbic homeostasis. They exert antidepressant, anxiolytic, and neuromodulatory effects (see more in Ref. [8]). In turn, rubiscolins are peptides isolated from spinach protein D-ribulose-1,5-bisphosphate carboxylase (RuBisCo); they act as G-protein-biased full agonists for δ-opioid receptor, as well as partial agonists for the µ/δ-opioid receptor heteromers [9,10,11]. Their chemical structure is quite distinct from other traditional opioid peptide structures; however, the presence of the protonated Tyr at the N-terminus, Pro in position 2 (as in endomorphins), and an aliphatic amino acid (Leu) in position 3 seems to be the minimal requirement for the opioid activity, as was soon confirmed in the structure–activity relationship studies [9,10]. Orally administered R-6 showed an antinociceptive, anxiolytic, and antidepressant effect, influencing learning and memory. Moreover, rubiscolins are promising in terms of their unique effects, such as stimulation of food intake or enhancement of glucose uptake in skeletal muscle [10,11].

The research results so far confirm that opioid peptides promote the survival of neurons during the development of the nervous system, influencing their proliferation and migration [5]. Endomorphins were considered cytoprotective and prosurvival agents; they scavenged radicals, inhibited lipid peroxidation, DNA, and protein oxidative damage [12], as well as prevented oxidative damage of LDL in in vitro injury models generated by 2,2′-azobis (2-amidinopropane hydrochloride) (AAPH), H_2_O_2_, or copper ions [13]. Endomorphins, in a concentration-dependent manner, inhibited copper ions and AAPH-induced changes due to the free radical scavenging effects. EM-1 was more potent than EM-2 and a major intracellular water-soluble antioxidant (L-glutathione). EM-2 was proposed as a lead compound for drug development in Alzheimer’s disease treatment and at the dose of 1–200 μM reduced the neurotoxicity of Aβ1–42 in differentiated SH-SY5Y cell line [14]. Tetrapeptide could bind to Aβ1–42 but did not arrest fibril formation or disassemble preformed aggregates.

Moreover, both endomorphins protected against intracellular Aβ toxicity in human and rat primary neuronal cultures, as well as in rat brains in vivo [15]. Cell death of neuronal cells was reduced to ~25%; additionally, endomorphins were able to protect mitochondrial function in an in vitro mouse brain mitochondria anoxia–reoxygenation model [16]. Mitochondrial respiratory activity improved, and oxidative-stress-induced changes in mitochondrial membrane fluidity, lipid peroxidation, and cardiolipin release were attenuated. Endomorphins also blocked the induced release of cytochrome c, which resulted in the inhibition of cell apoptosis. Regarding rubiscolins, there is no evidence of their neuroprotective potential, raising the question of whether rubiscolins, with their anxiolytic-like, antidepressant-like, and memory-enhancing activities, can counteract neuron injury. Thus, the main goal of this work is to characterize the therapeutic potential of endomorphins and rubiscolins against neurotoxicity induced by 6-OHDA in SH-SY5Y neuronal cells.

## 2. Results

### 2.1. Peptide Synthesis

Peptides were synthesized by standard solid-phase procedures as described before [17] using techniques for 9-fluorenylmethoxycarbonyl (Fmoc)-protected amino acids on Fmoc-Phe Wang (100–200 mesh, 0.69 mM/g, Novabiochem) or MBHA Rink-Amide peptide resin (100–200 mesh, 0.80 mM/g, Novabiochem) and 2-(1H-benzotriazol-1-yl)-1,1,3,3-tetramethyluronium tetrafluoroborate (TBTU) as a coupling agent. The final purity of all peptides was >98%. Calculated values for protonated molecular ions were in agreement with those determined by FAB mass spectrometry. The physicochemical data of tested peptides are presented in Table 1.

### 2.2. Antioxidant Activity of Peptides

The antioxidant activity of endomorphins and rubiscolins was evaluated by the means of different chemical methods, namely DPPH radical scavenging, ferric-reducing antioxidant power assay (FRAP) and oxygen radical absorption capacity (ORAC). Three independent experiments of each method were carried out with *n* = 3 and the results obtained are shown in Figure 2A–C. For FRAP and ORAC assay, the values were extrapolated by the means of calibration curve using as standard FeSO_4_ and Trolox, respectively. The detailed data on antioxidant capacity are provided in the Appendix A.

Neither endomorphins nor rubiscolins displayed the ability to reduce the DPPH radical when compared to the control (Figure 1A), even if tested at three different concentrations (Appendix A). Similar results for R-6 were obtained by Stefanucci et al. [18]. In FRAP assay (Figure 1B), it is possible to observe that R-6 (1.10 ± 0.20 µM Eq. FeSO_4_/g of peptide) exhibited the highest ability to reduce iron ions. What is more, EM-1 (9115.0 ± 112.9 µmol Eq. Trolox/g of peptide) displayed the highest values of ORAC (Figure 1C), while other peptides showed three times less ability to neutralize the peroxyl radicals. Antioxidant activities of peptides and standards are provided in the Appendix A.

### 2.3. Cytotoxicity of Peptides on SH-SY5Y Cells

The cytotoxic effects of peptides on SH-SY5Y cell viability were evaluated and the results are presented in Figure 3.

The results showed that endomorphins and R-5 at the dose of 1–10 µM did not induce cytotoxic effects on SH-SY5Y cells. Only R-6 at the highest dose (10 µM) significantly decreased the viability of the SH-SY5Y cells by more than 20%. 

### 2.4. Neuroprotective Effects of Peptides on SH-SY5Y Cells

The neuroprotective effects of peptides (0.1–10 µM; 24 h) were evaluated in the presence of the 6-OHDA neurotoxin (Figure 4) at sub-toxic concentrations.

The exposure of SH-SY5Y cells to 6-OHDA (100 µM) for 24 h reduced the cell viability by about 52% (48.63 ± 1.25% of viable cells) when compared to vehicle (100.00 ± 4.60% of viable cells). However, SH-SY5Y cells treated with 6-OHDA in the presence of peptides (0.1–10 µM) exhibited a significant capacity to recover the neurotoxicity induced by 6-OHDA treatment. Rubiscolin-6 prevented cell death in 20% at the lowest dose, while R-5 was inactive. Endomorphins also significantly revealed capacity (7–10%) to recover the neurotoxicity induced by 6-OHDA treatment.

The LDH, as an intracellular marker enzyme, has a stable chemical and biological property; thus, the amount of LDH in the cytosol is always taken as an indicator for cell damage and cell membrane permeability. The LDH activity was determined using a commercially available LDH Activity Assay kit based on an enzymatic coupling reaction in which LDH reduces NAD to NADH, which is detected by colorimetric assay. 

The results showed (Figure 5) that the exposure of SH-SY5Y cells to 6-OHDA (100 µM) for 24 h significantly increased LDH activity (by about 50%) when compared to vehicle. Endomorphin-1 and R-6 significantly decreased LDH activity at the two lowest doses, which indicated that peptides limited the loss of membrane integrity and the presence of cell damage. Endomorphin-2 at the dose of 1 µM could also inhibit toxin-induced LDH activity. 

#### Neuroprotective Effects of Peptides and Opioid Receptor Antagonists on SH-SY5Y Cells

To study the possible involvement of opioid receptors in neuroprotective effect, EMs and R-6 at the two most active doses were coadministrated with the opioid receptor antagonists. SH-SY5Y cells were treated with β-funaltrexamine (β-FNA, µ-opioid receptor antagonist, 10 µM), naltrindole (NLT, δ-opioid receptor antagonist, 10 µM), and nor-binaltorphimine (NOR-BNI, κ-opioid receptor, 10 µM) 30 min before peptides and 6-OHDA (100 µM). Cell viability was assessed after treatment by the MTT assay.

Concurrent exposure of cells to peptides and opioid receptor antagonists did not attenuate the peptide neuroprotection (Figure 6); however, changes are not significant. In the case of endomorphins a slight µ-opioid receptor antagonist effect is evident. The neuroprotective effect of R-6 was not abolished by the administration of antagonists, so we can assume that a different mechanism may be responsible for the observed effects.

### 2.5. Effects of Peptides on Hallmarks Related with PD

Several hallmarks associated with neuronal cell death on PD were evaluated, namely ROS production, MMP, and Caspase-3 activity. These hallmarks were evaluated on SH-SY5Y cells treated with 6-OHDA in the presence and absence of peptides (0.1–10 µM), and the results are presented in Figure 7, Figure 8 and Figure 9, respectively. 

#### 2.5.1. Production of Reactive Oxygen Species (ROS)

The intracellular ROS levels were determined in control, 6-OHDA-injured (100 µM), and 6-OHDA-injured cells treated with different concentrations of peptides. The exposure of cells to 6-OHDA led to a marked increase in ROS levels by around 80% compared with vehicle (Figure 7). Endomorphins could not markedly decrease the ROS production stimulated by 6-OHDA treatment (only 20%). However, R-6 at the dose of 0.1 µM displayed the ability to significantly reduce the ROS levels by around 70% compared with 6-OHDA.

#### 2.5.2. Mitochondrial Membrane Potential

To understand if the neuroprotective effect of peptides was mediated by biological events on mitochondria, the MMP levels were evaluated. The exposure of SH-SY5Y cells to 6-OHDA (100 µM) led to a marked increase in MMP depolarization (+114%) when compared to vehicle (Figure 8). In the presence of endomorphins, only EM-1 at the dose of 1 µM showed a significant decrease in MMP depolarization induced by 6-OHDA treatment (by about 64%). Plant-derived R-6 also markedly ameliorated the changes caused by the neurotoxin. It was possible to observe a noticeable inhibitory change caused by peptide at the dose of 0.1 µM (by about 73%) and 1 µM (by about 50%) in the depolarization induced by neurotoxin exposition.

#### 2.5.3. Caspase-3 Activity 

The exposure of cells to 6-OHDA (100 µM) led to a marked increase in Caspase-3 activity (5.22 ± 0.38 Δ fluorescence (a.u.)/mg of protein/min) compared to vehicle (1.21 ± 0.08 Δ fluorescence (a.u.)/mg of protein/min) (Figure 9). In the presence of EM-2, a significant decrease in Caspase-3 activity, of 1.74–1.79 Δ fluorescence (a.u.)/mg of protein/min, was observed, while EM-1 and R-6 at a dose of 1, 0.1, and 1 µM, respectively, did not decline the Caspase-3 activity when compared to vehicle.

### 2.6. Estimation of BDNF Using ELISA

The BDNF molecule is a member of the neurotrophin family of secreted proteins that plays a critical role in neuronal development and synaptic plasticity. This signaling molecule has a crucial role in the survival and differentiation of neuronal populations during development; it can regulate the synthesis of synaptic proteins and elevate synaptic efficacy, necessary precursors for appropriate neuronal function, survival, and apoptosis [19]. Since we know that BDNF expression has been associated with both normal and pathological aging, psychiatric disease, and cognitive and neurodegenerative diseases, we evaluated the BDNF protein level in the injured SH-SY5Y cell line using ELISA assay.

The protein levels of BDNF w significantly reduced after 24-h incubation with 100 μM 6-OHDA (Figure 10). Exposure of injured cells to EM-2 did not increase the BDNF level in a significant manner. Only R-6 at the dose of 0.1 μM was able to significantly stimulate BDNF level as compared with injured cells, which may represent a possible mechanism by which R-6 mediated its neuroprotective potency.

## 3. Discussion

To prevent or mitigate the effects of neurodegeneration processes, the administration of neuroprotective agents may be beneficial. Candidates for a neuroprotective drug can work in various ways to directly or indirectly reduce neuron damage, mitochondrial dysfunction, oxidative stress or neuroinflammation. Common synthetic drugs do not alter the disease course substantially enough, and either are associated with side effects or lose their effectiveness after extended use. The search for new compounds with potential health benefits is increasingly essential, especially in the case of diseases with alarmingly growing rates, such as neurodegenerative diseases. Consequently, a class of proteins and peptides have shown promising effects because they are extraordinary biomolecules characterized by a broad spectrum of activities and the ability to act through diverse mechanisms of action. Some of the neuroprotective peptides can prevent oxidation by the same processes as well-known natural antioxidants [20]; they can inhibit oxidative stress factors, quench excess free radicals, or stimulate the activities of antioxidant enzymes [21]. The molecular weight, peptide sequence, and amino acid composition all determine the peptides’ antioxidant properties. Moreover, peptides containing amino acid residues, such as Asp, Pro, Trp, Tyr, Met, Cys, Leu, Arg, Ala, Phe, and His, showed higher antioxidant activities [22]. There is some evidence that hydrophobic amino acids, such as Ala, Leu, Met, or Val (particularly at the N-termini), and aromatic Phe, Tyr, Trp, and/or the imidazole ring-containing His (in the sequence) can enhance the radical-scavenging abilities of peptides [20,22]. Amino acids with aromatic residues donate protons to electron-deficient radicals [20], while nucleophilic sulfur-containing amino acid residues (Cys, Met) interact with radicals, e.g., the SH group in Cys, as a radical scavenger, protects tissue from oxidative stress and improves the glutathione activity [22]. Moreover, carboxyl and amino groups in the side chains of amino acids (Asp, Glu, and His) serve as a chelator of metal ions [23] and hydrogen donors [24]. 

Generally, peptides can exert numerous biological features that make them valuable molecules to apply as potential therapeutic agents to improve health benefits, especially in neuroprotection. The present study investigated opioid peptides’ effect against 6-OHDA-induced neurotoxicity in SH-SY5Y cell lines that are widely used as a cellular model for PD research. These cells possess many characteristics of dopaminergic neurons and they express tyrosine hydroxylase and dopamine-β-hydroxylase as well as the dopamine transporter, even without being differentiated [25]. Our results showed that 6-OHDA reduced cell viability, increased ROS production and mitochondrial membrane depolarization, and stimulated Caspase-3 activity compared with untreated SH-SY5Y cells. This experiment showed that endomorphins and R-6 treatment significantly attenuated changes in cell viability induced by the neurotoxin, while the exposition of injured cells to R-5 did not reverse cell impairment. The loss of cell viability goes along with several hallmarks related to cell death typical in neurodegeneration. The increase in ROS levels caused by 6-OHDA has harmful effects on cell homeostasis, structures, and functions and results in oxidative stress. These unfavorable changes may amplify the apoptotic signaling pathway and, thus, lead to cell death. Endomorphins only slightly decreased ROS production (~20%). However, R-6 at the dose of 0.1 µM significantly alleviated the ROS induced by 6-OHDA (~70%), which agrees with the ferric-reducing antioxidant power of R-6.

The only difference between rubiscolins’ structure concerns the presence of Phe in position 6 of R-6 sequence. As mentioned above, aromatic Phe can enhance the radical scavenging abilities of peptides [20,22]. These results suggest that the hypothetical underlying mechanism mediating R-6 neuroprotection may be involved in the protection against oxidative stress. Additionally, R-6 markedly ameliorates the changes in MMP caused by the neurotoxin. However, plant-derived peptides stimulated the Caspase-3 activity coinciding with the neuroprotective period, without subsequent cell death in vitro. It is appealing to speculate that Caspase-3 activation is critical for the determination of cell fate after injury. Activation of caspases is recognized as occurring downstream of the commitment to die decision. However, emerging evidence now suggests that multiple factors may protect against apoptosome assembly and cleaved caspases [26]. Generally, we speculated that peptides’ ability to suppress the toxic effect induced by 6-OHDA may be mediated by different cellular mechanisms. The protective effect caused by EM-2 results in an antiapoptotic effect (mitochondrial protection and decrease in Caspase-3 activity), while R-6 potency increasing cells’ viability seems to be mediated by reducing oxidative stress. Additionally, in our study, the protein level of BDNF was increased significantly after 24-h incubation with 0.1 μM R-6; this may represent a compensatory response to decreased BDNF levels in SH-SY5Y cells following 6-OHDA exposure. BDNF is a well-studied key factor in brain development, including neuronal cell survival, differentiation, migration, and plasticity. Till now, several natural or synthetic compounds exert neuroprotective effects by enhancing the expression of BDNF, e.g., citrus flavonoid 3,5,6,7,8,30,40-heptamethoxyflavone (HMF) [27], or nutraceuticals, such as Phlorotanin extract and β-carotene [28]. Several studies reported peptides displaying the prosurvival and neuroprotective effects associated with inhibition of oxidative stress, apoptosis, and inflammation [5]. Among them, there are antioxidant peptides (e.g., Pro-Ala-Tyr-Cys-Ser, Cys-Val-Gly-Ser-Tyr [29], Dmt-D-Arg-Phe-Lys-NH_2_, Phe-D-Arg-Phe-Lys-NH_2_, and D-Arg-Dmt-Lys-Phe-NH_2_ [30,31]), mitochondria-targeted peptides (e.g., cationic-arginine-rich peptides, Penetratin [32], neuroprotective peptides fused to cell-penetrating peptides [33,34]), endogenous peptides (e.g., Apelin-family peptides [35] and Humanin [36]), opioid peptides (e.g., mentioned above endomorphins [12,13,14,15,16] and Tyr-D-Ala-Gly-MePhe-Gly-ol [37]), and many others (see more in Ref. [5]) considered as a novel therapeutic agent against various neurodegenerative diseases. Based on these recent studies, we hypothesize that peptides could be a good candidate for therapeutic agents to prevent neuronal degeneration and loss of function. The data obtained suggest that the protective effect induced by endomorphins results in mitochondrial protection and reduction in Caspase-3 activity, while R-6 potency increasing cells’ viability seems to be mediated by reducing oxidative stress. These naturally occurring peptides exhibited prosurvival potency, being interesting candidates for further neuroprotective studies. Despite the results attained, further studies should be accomplished in more complex in vitro cellular models, such as differentiated cells, cocultures, and/or 3D cellular models to understand the neuroprotective potential of peptides.

## 4. Materials and Methods

### 4.1. Peptide Synthesis

Peptides were synthesized by routinely used standard solid-phase procedures in the Department of Biomolecular Chemistry (Medical University of Lodz, Poland) as described before [17]. Convenient and efficient techniques for 9-fluorenylmethoxycarbonyl (Fmoc)-protected amino acids on Fmoc-Phe Wang (100–200 mesh, 0.69 mM/g, Novabiochem, La Jolla, CA, USA) or MBHA Rink-Amide peptide resin (100–200 mesh, 0.80 mM/g, Novabiochem) and 2-(1H-benzotriazol-1-yl)-1,1,3,3-tetramethyluronium tetrafluoroborate (TBTU) as a coupling agent have been used. Crude peptides were purified by preparative reversed-phase HPLC on a Vydac C18 column (10 μm, 22 × 250 mm) equipped with a Vydac guard cartridge. For purification, a linear gradient of 0–100% acetonitrile containing 0.1% TFA over 15 min at a flow rate of 15 mL/min was used. The purity of the final peptides was verified by analytical HPLC employing a Vydac C18 column (5 μm, 4.6 × 250 mm) and the solvent system of 0.1% TFA in water (A) and 80% acetonitrile in water containing 0.1% TFA (B). A linear gradient of 0–100% solvent B over 25 min at a flow rate of 1 mL/min was used for the analysis. The absorbance was monitored at 214 nm. The final purity of synthesised peptides was >98%. Calculated values for protonated molecular ions were in agreement with those determined by FAB mass spectrometry. Table 1 showed the physicochemical data of tested peptides.

### 4.2. Antioxidant Activity

#### 4.2.1. DPPH (1,1-Diphenyl-2-picrylhydrazyl) Radical Scavenging Activity

DPPH radical scavenging activity was performed according to Silva et al. [38]. Briefly, 2 µL of sample solution (10 µM) was added to 198 µL of the DPPH radical solution (0.1 mM) and the mixture was incubated in the dark for 30 min at room temperature. The absorbance was read at 517 nm. Butylated hydroxytoluene (BHT) was used as a standard. 

#### 4.2.2. Oxygen Radical Absorption Capacity Method (ORAC)

This method evaluated the ability of peptides to neutralize the peroxyl radicals according to Dávalos et al. [39] with slight modifications. Sample (20 µL) and fluorescein (120 µL; 70 nM, final concentration) were placed in a microplate well and preincubated for 15 min at 37 °C. A total of 60 µL of AAPH (2,2′-azobis(2-amidino-propane) dihydrochloride), a radical generator, was added to a final concentration of 12 mM. The microplate was immediately placed in the reader, and the fluorescence (λexcitation: 458 nm; λemission: 520 nm) was recorded every minute for 240 min. Trolox (0–80 µM), as antioxidant standard, was also carried out in each assay. Final results were expressed in µmol of Trolox equivalents/µM peptide (µmol TE/µM peptide).

#### 4.2.3. Ferric-Reducing Antioxidant Power (FRAP)

The principle of FRAP assay was based on reducing Fe^3+^ complex to the ferrous form (Fe^2+^). This method [40] was evaluated according to Silva et al. [38]. FeSO_4_ was used as standard. FRAP reagent was prepared with 0.3 M Acetate Buffer (pH = 3.6), 10 mM of TPTZ in 40 mM HCL, and 20 mM ferric solution using FeCl_3_ by freshly mixing acetate buffer, TPTZ, and ferric solutions at a ratio of 10:1:1. The final working FRAP reagent was then incubated at 37 °C. Briefly, 2 µL of the sample was added to 198 µL of FRAP reagent and allowed to stand at 37 °C in the dark for 30 min, at which time the absorbance was measured at 593 nm using a microplate reader. The difference between the absorbance of the test sample and the blank reading was calculated and expressed as mM of FeSO_4_ per mg of the peptide.

### 4.3. Neuroprotective Activity of Peptides on SH-SY5Y Cells

#### 4.3.1. Cell Culture Maintenance 

The neuroblastoma cell line (SH-SY5Y) was obtained from the German Collection of Microorganisms and Cell Cultures GmbH (DSMZ) Bank (ACC 209). Cells were cultivated at 37 °C and 5% CO_2_ with DMEM:F12 medium containing 1% antibiotic/antimycotic (amphotericin B, penicillin, and streptomycin) (Biowest, Nuaillé, France) and 10% (*v*/*v*) fetal bovine serum (FBS) (Biowest, Riverside, MO, USA).

#### 4.3.2. Cytotoxic and Neuroprotective Activities of Peptides 

Effects on cell viability and neuroprotection assays were estimated using the MTT (VWR, Solon, OH, USA) method, as described by Silva et al. [38]. SH-SY5Y cells were exposed for 24 h to 6-OHDA (100 µM) in the absence/presence of peptides at nontoxic concentrations (0.1–10 µM). Then, 100 µL MTT (1.2 mM) was added to wells, and cells were incubated for 1 h at 37 °C. After this time, MTT was removed and 100 µL DMSO was added. The resulting absorbance was read in a microplate reader (Bioteck, Epoch/2 microplate reader, Winooski, VT, USA) at 570 nm. The results were expressed in the percentage of control. 

Additionally, a commercially available LDH Activity Assay kit (Sigma-Aldrich, St. Louis, MI, USA) was used to evaluate the activity of lactate dehydrogenase (LDH) in the SH-SY5Y cell culture medium according to the manufacturers’ instructions. Briefly, SH-SY5Y cells were exposed for 24 h to 6-OHDA (100 µM) in the absence/presence of peptides at nontoxic concentrations (0.1–10 µM). At the end of the incubation time, the cells were centrifuged at 1300× *g* for 10 min (MPW-350R, MPW Med. Instruments, Warsaw, Poland). Then, 25 µL of the sample was incubated with LDH Assay Buffer and Master Reaction Mix in the dark at 37 °C. Absorbance was read at 450 nm at 37 °C using a microplate reader (FlexStation 3 Multi-Mode Microplate Reader, Molecular Devices, San Jose, CA, USA). Measurements were taken every 5 min, until the value of the most active sample was greater than the value of the highest standard (12.5 nmole/well). The LDH activity of a sample was determined by the following Equation:$$\mathrm{LDH}\;\mathrm{Activity}=\frac{\mathrm{Amount}\;\left(\mathrm{nmole}\right){\;\mathrm{of}\;\mathrm{NADH}\;\mathrm{generated}\;\mathrm{between}\;T}_{\mathrm{initial}\;}{\;\mathrm{and}\;T}_{\mathrm{final}}}{T_{\mathrm{final}}-{\;T}_{\mathrm{initial}}\;\left(\mathrm{minutes}\right)\times \;V}\times \mathrm{Sample}\;\mathrm{Dilution}\;\mathrm{Factor}$$

At the end, the percentage of cytotoxicity was calculated based on the 100% viability of nontreated cells. 

To determine if opioid receptors were involved in the neuroprotective effect of EMs and R-6, SH-SY5Y cells were treated with β-FNA (µ-opioid receptor antagonist, 10 µM) and/or NLT (δ-opioid receptor antagonist, 10 µM), NOR-BNI (κ-opioid receptor antagonist, 10 µM) 30 min before peptides, and 6-OHDA. Cell viability was assessed after treatment by the MTT assay.

### 4.4. Study of Intracellular Signaling Pathways 

#### 4.4.1. ROS Production

ROS levels were evaluated using the 5(6)-carboxy-20,70- dichlorofluorescein diacetate (carboxy-H2DCFDA) probe (Invitrogen, Bleiswijk, The Netherlands) [38]. After incubation with peptides, cells were washed with ice-cold phosphate-buffered saline (PBS) and C-H2DCFDA (100 µL, 20 µM) was added, and they wereincubated for 1 h at 37 °C. The resulting fluorescence was read at 527 nm excitation and 590 nm emission wavelengths, and ROS levels were presented in the percentage of control (nontreated cells).

#### 4.4.2. Mitochondrial Membrane Potential (∆Ψm)

MMP was evaluated according to Silva et al. [38] using JC-1 probe (Molecular Probes, Eugene, OR, USA). SH-SY5Y cells were exposed for 6 h to 6-OHDA (100 µM) in the presence/absence of peptides. Cells were washed with ice-cold PBS, and then 200 µL of JC-1 (3 µM) was added prior to 15 min at 37 °C. Afterwards, JC-1 was removed and PBS was added. The monomers/aggregates formation was determined by fluorescence extrapolation at 530 nm emission (monomers)/590 nm (aggregates) and 490 nm excitation wavelengths for 30 min at 37 °C. MMP was calculated through the ratio between monomers/aggregates formation and presented as a percentage of control.

#### 4.4.3. Caspase-3 Activity

Enzyme activity was evaluated according to Silva et al. [4]. SH-SY5Y cells were exposed for 6 h to 6-OHDA (100 µM) in the presence/absence of peptides. Then, the cells were rinsed with ice-cold PBS, scraped, and centrifuged at 3300× *g* for 5 min. SH-SY5Y cells were then incubated with lysis buffer on ice for 20 min and, finally, centrifuged at 22,500× *g*, 4 °C, for 20 min. Cell lysates were processed following the manufacturer’s protocol “Caspase-3 fluorometric assay” (Sigma, St. Louis, MO, USA) and fluorescence was read at 360 nm excitation and 460 nm emission wavelengths. Caspase-3 activity was calculated through the curve slope and presented as Δ fluorescence (a.u.)/mg of protein/min).

### 4.5. Estimation of Brain-Derived Neurotrophic Factor (BDNF) Using ELISA

After 24-h exposure to 6-OHDA (100 µM M) and EM-2 (1 and 10 µM) or R-6 (0.1 and 1 µM), the supernatants of SH-SY5Y cells cultured in 6-well plates were collected and centrifuged at 150× *g* at 4 °C to remove floating cells, and then the prepared supernatants were immediately added to a 96-well plate coated with a primary antibody specific to human BDNF according to the manufacturers’ instructions (Human BDNF SimpleStep ELISA^®^ Kit, Abcam, Cambridge, Great Britain). Briefly, 50 μL of all samples or standard was added to appropriate wells, together with 50 μL of the Antibody Cocktail. The plate was sealed and incubated for 1 h at room temperature on a plate shaker set to 400 rpm. Each well was then washed three times with 250 μL 1× Wash Buffer PT. Then, TMB substrate solution as the developing agent was added to each well and incubated for 10 min in the dark on a plate shaker set to 400 rpm, and color developed in proportion to the amount of bound BDNF. Finally, the stop solution changed the color from blue to yellow, and the intensity was measured at 450 nm. A standard curve was run for each assay, and all standards or samples were run in triplicate. 

### 4.6. Data and Statistical Analysis

At least three or four independent experiments were carried out in triplicate and results were presented as mean ± standard error of the mean (SEM). ANOVA with Dunnett’s multiple comparisons of group means analysis was accomplished and the Tukey’s test was applied for multiple comparisons. Differences were considered significant at the level of 0.05 (*p* < 0.05). GraphPad v8.0 (GraphPad Software, La Jolla, CA, USA) software was used to accomplish the analysis.

## 5. Conclusions

The present study discloses the therapeutic potential of naturally occurring peptides in 6-OHDA-induced neurotoxicity in an in vitro model of PD. Our findings suggest that R-6 exerts protective effects, possibly related to an antioxidation mechanism and an ability to increase BDNF, while endomorphins prevent apoptosis. Further studies should be accomplished in more relevant cellular or animal models to understand the exact mechanism of neuroprotection. Overall, the findings herein provide novel targets for developing PD therapies and a theoretical basis of peptides for future clinical applications.

## Figures and Tables

**Figure 1 ijms-23-11778-f001:**
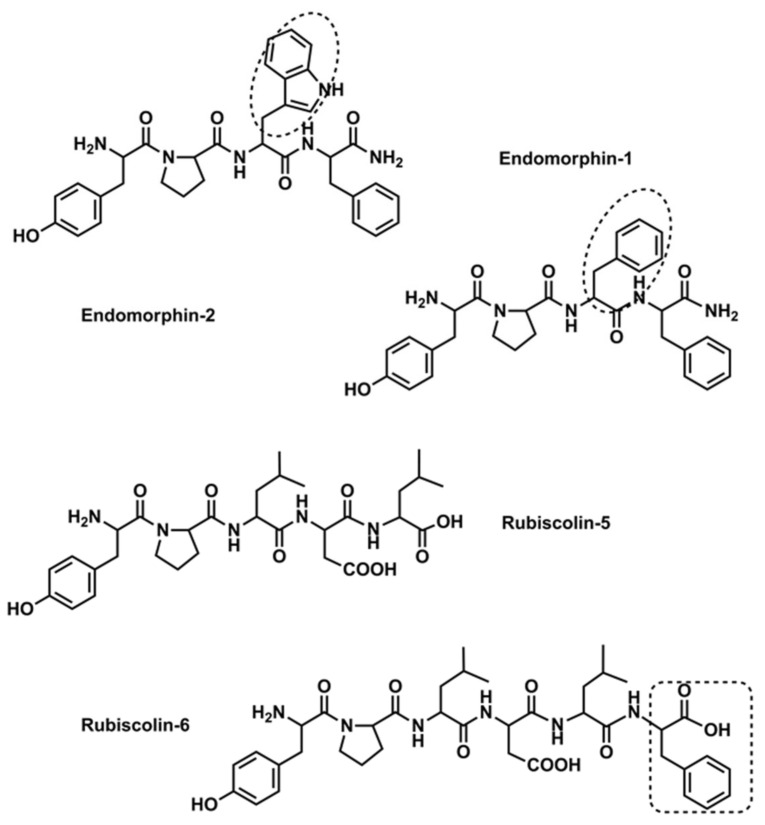
Endomorphin and rubiscolin structures (the differences in chemical structure are indicated with dotted lines).

**Figure 2 ijms-23-11778-f002:**
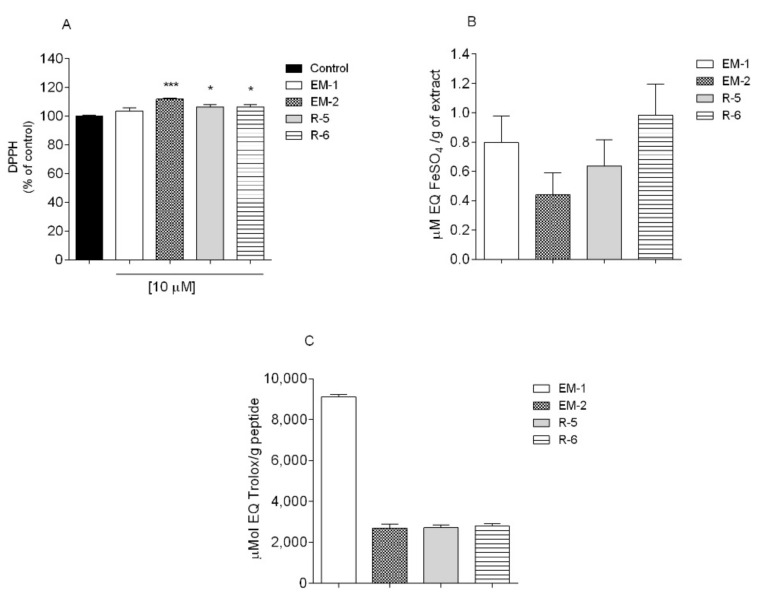
The antioxidant activity of the peptides evaluated through the DPPH (**A**), FRAP (**B**), and ORAC (**C**) methods. The values in each column represent the mean ± standard error of the mean (SEM) of 3 or 4 in-dependent experiments. Symbols represent significant differences (ANOVA, Dunett’s test, * *p* < 0.05, *** *p* < 0.001) when compared to: control.

**Figure 3 ijms-23-11778-f003:**
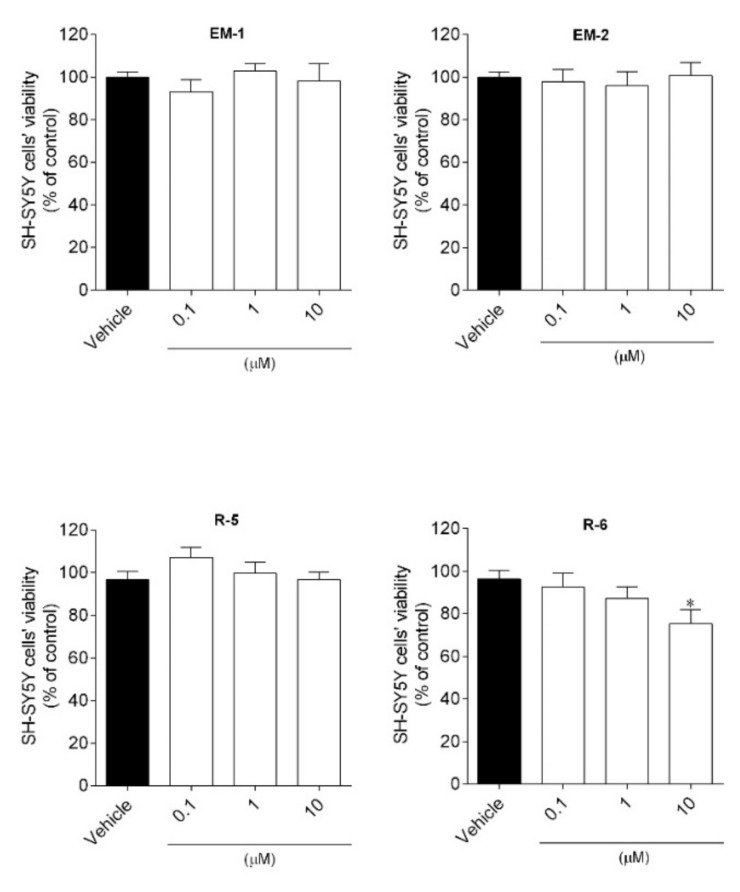
SH-SY5Y cells’ viability when exposed 24 h to peptides (0.1–10 µM). The values in each column represent the mean ± SEM of 3 or 4 independent experiments. Symbols represent significant differences (ANOVA, Dunett’s test, * *p* < 0.05) when compared to: vehicle.

**Figure 4 ijms-23-11778-f004:**
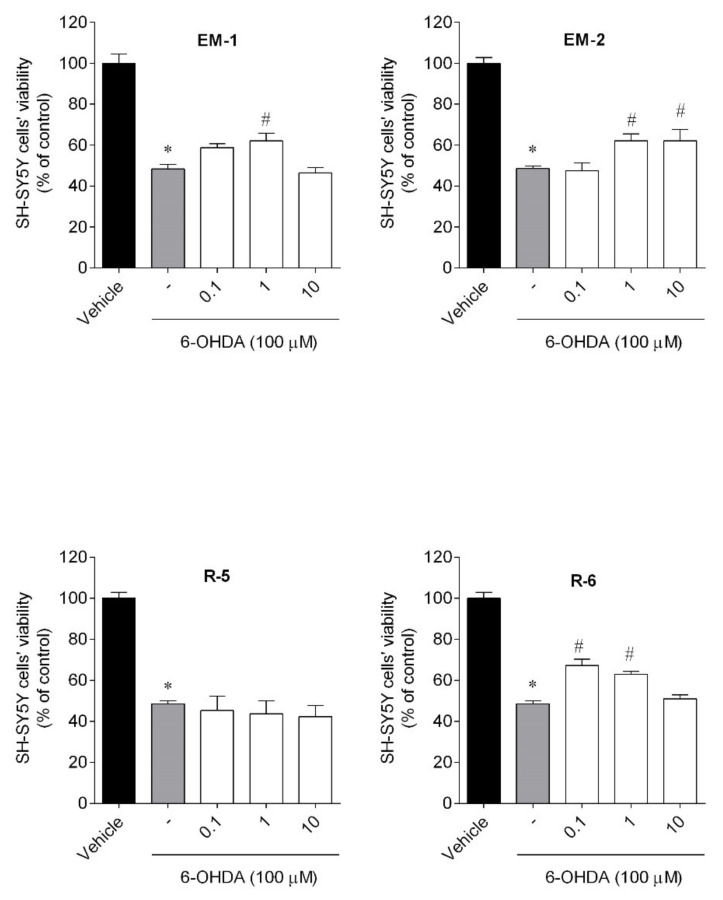
Neuroprotective effects on SH-SY5Y cells treated with 6-OHDA (100 µM) in the presence/absence of peptides (0.1–10 µM, 24 h). The values in each column represent the mean ± SEM of 3 or 4 independent experiments. Symbols represent significant differences (ANOVA, Dunnett’s test, *p* < 0.05) when compared to: * vehicle and # 6-OHDA. (-) 6-OHDA.

**Figure 5 ijms-23-11778-f005:**
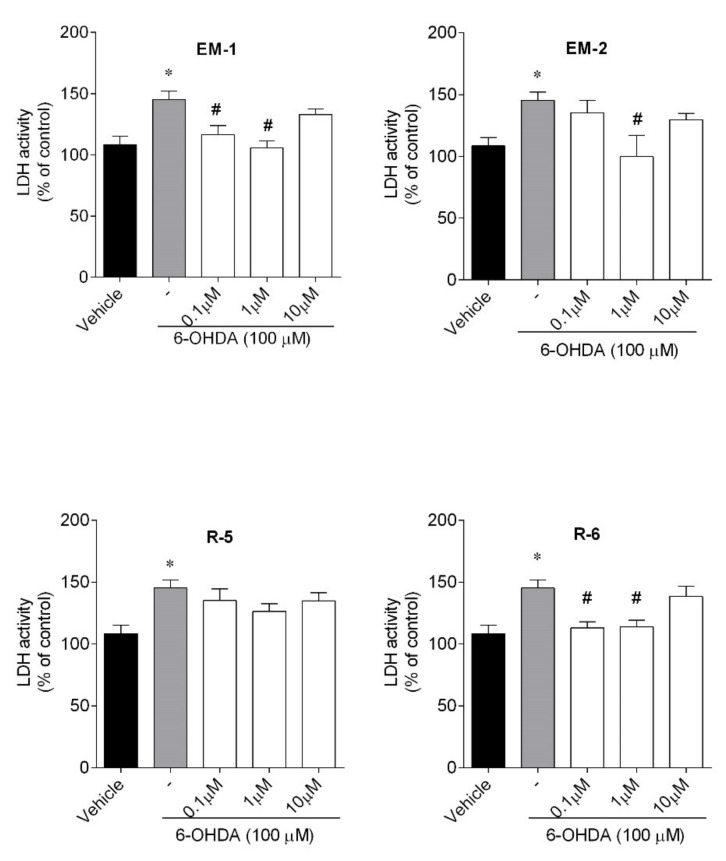
Effect of peptides (0.1–10 µM, 24 h) on LDH activity in 6-OHDA-treated (100 µM) SH-SY5Y cells. The values in each column represent the mean ± SEM of 4 independent experiments. Symbols represent significant differences (ANOVA, Dunnett’s test, *p* < 0.05) when compared to: * vehicle and # 6-OHDA. (-) 6-OHDA.

**Figure 6 ijms-23-11778-f006:**
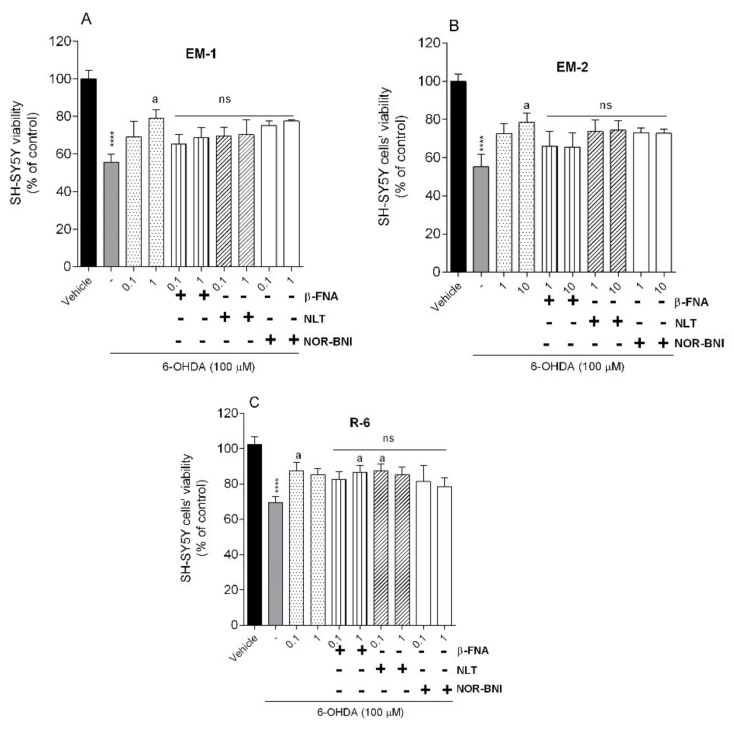
Neuroprotective effects on SH-SY5Y cells treated with 6-OHDA (100 µM) in the presence of (**A**) EM-1, (**B**) EM-2, (**C**) R-6, respectively, and β-FNA (µ-opioid receptor antagonist, 10 µM), NLT (δ-opioid receptor antagonist, 10 µM), or NOR-BNI (κ-opioid receptor, 10 µM). The values in each column represent the mean ± SEM of 3 independent experiments. **** *p* < 0.0001, as compared to vehicle using one–way ANOVA, followed by the Tukey test; ^a^
*p* < 0.05, as compared to 6-OHDA; ns (not significant), as compared to peptides only. (-) 6-OHDA.

**Figure 7 ijms-23-11778-f007:**
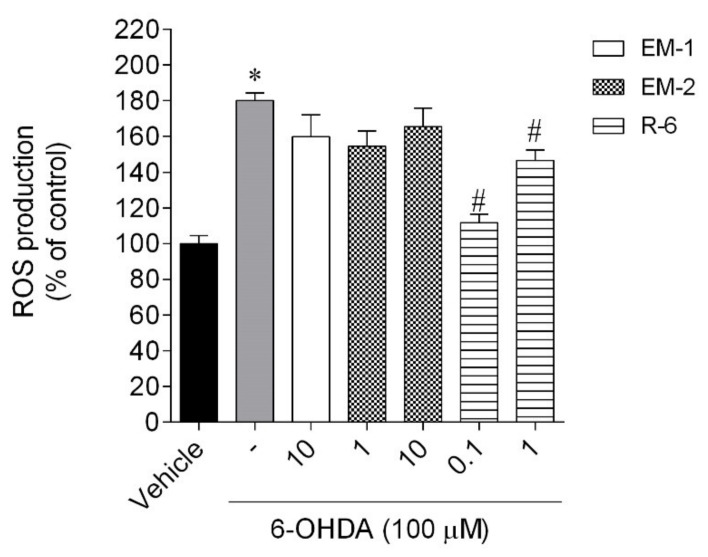
ROS production of SH-SY5Y cells treated with 6-OHDA in the presence/absence of peptides (0.1–10 µM, 6 h). The values in each column represent the mean ± standard error of the mean (SEM) of 3 or 4 independent experiments. Symbols represent significant differences (ANOVA, Dunnett’s test, *p* < 0.05) when compared to: * vehicle and # 6-OHDA. (-) 6-OHDA.

**Figure 8 ijms-23-11778-f008:**
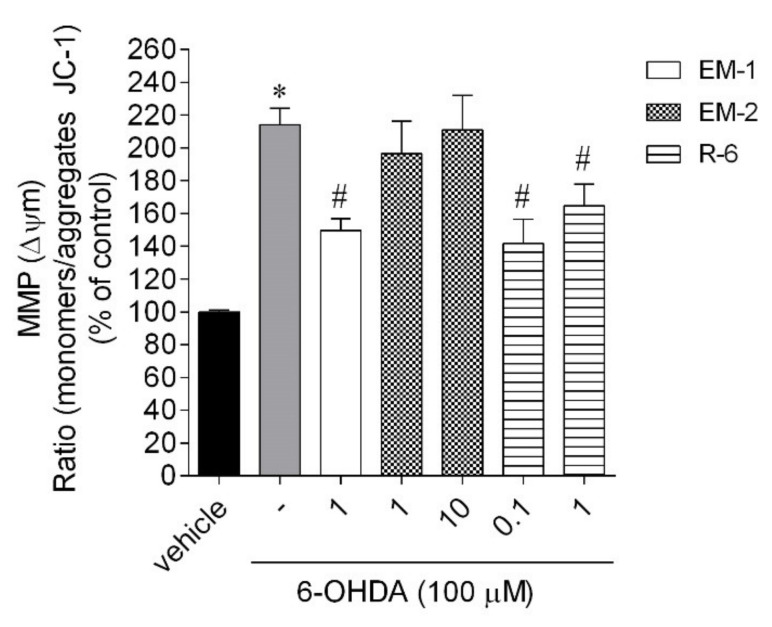
Changes in mitochondrial membrane potential (MMP) of SH-SY5Y cells following treatment with 6-OHDA (100 µM) in the presence/absence of peptides (0.1–10 µM, 6 h). The values in each column represent the mean ± standard error of the mean (SEM) of 3 or 4 independent experiments. Symbols represent significant differences (ANOVA, Dunnett’s test, *p* < 0.05) when compared to: * vehicle and # 6-OHDA. (-) 6-OHDA.

**Figure 9 ijms-23-11778-f009:**
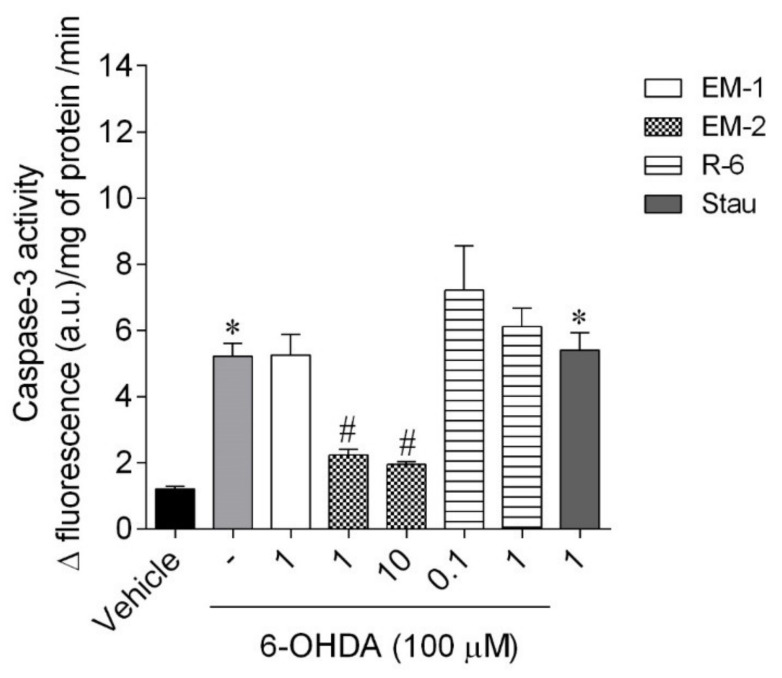
Caspase-3 activity of SH-SY5Y cells treated with 6-OHDA in the presence/absence of peptides (0.1–10 µM, 6 h). The values in each column represent the mean ± standard error of the mean (SEM) of 3 or 4 independent experiments. Symbols represent significant differences (ANOVA, Dunnett’s test, *p* < 0.05) when compared to: * vehicle and # 6-OHDA. (-) 6-OHDA.

**Figure 10 ijms-23-11778-f010:**
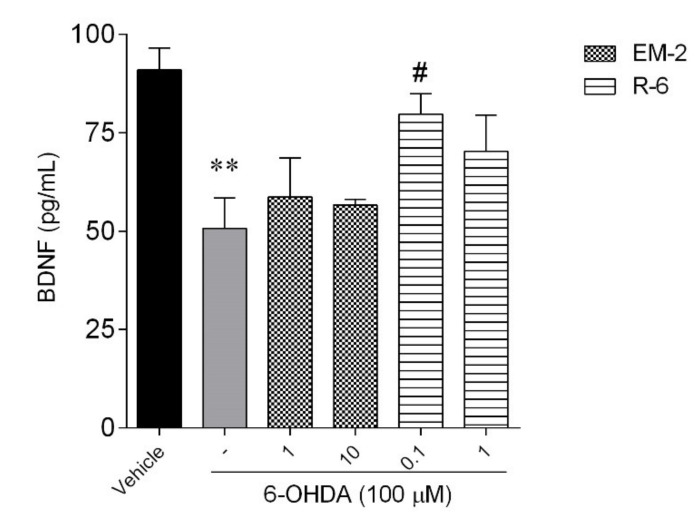
BDNF protein levels in the culture supernatant from SH-SY5Y cells treated by 6-OHDA (100 µM) and EM-2 (1 and 10 µM) or R-6 (0.1 and 1 µM). The values in each column represent the mean ± standard error of the mean (SEM) of 3 independent experiments. Symbols represent significant differences (ANOVA, Dunnett’s test): ** *p* < 0.01 when compared to vehicle and # *p* < 0.05 when compared to 6-OHDA. (-) 6-OHDA.

**Table 1 ijms-23-11778-t001:** Physicochemical data of endomorphins and rubiscolins.

Sequence	HPLC ^a^ (t_r_)	FABS-MS			Purity (%)
		Formula	MW	[M+H]^+^	
Tyr-Pro-Trp-Phe-NH_2_ (EM-1) *	17.29	C_34_H_37_N_6_O_5_	610	611	98
Tyr-Pro-Phe-Phe-NH_2_ (EM-2) *	16.81	C_34_H_37_N_6_O_5_	571	572	99
Tyr-Pro-Leu-Asp-Leu-OH (R-5)	16.78	C_30_H_45_N_6_O_9_	620	621	98
Tyr-Pro-Leu-Asp-Leu-Phe-OH (R-6)	16.31	C_39_H_54_N_6_O_10_	767	768	98

^a^ HPLC elution on a Vydac C18 column (5 µm, 4.6 × 250 mm) using the solvent system of 0.1% TFA in water (A)/80% acetonitrile in water containing 0.1% TFA (B) and a linear gradient of 0–100% solvent B over 25 min at a flow rate of 1 mL/min. * Data from Ref. [17].

## Data Availability

Not applicable.

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
