# Peer review of "The Therapeutic Potential of Naturally Occurring Peptides in Counteracting SH-SY5Y Cells Injury"

_ijms, 2022, doi:10.3390/ijms231911778_

Round 1

Reviewer 1 Report

The article by Renata Perlikowska et al. entitled “The therapeutic potential of naturally occurring peptides in counteracting SH-SY5Y cells injury” was submitted to IJMS. The reviewer's enthusiasm remains limited due to the following concerns.

1. The authors have not followed the preparation of the manuscript according to the author’s instructions. 

2. The author must write the aim, objectives, and conclusion of the study in the abstract.

3. Line 35-38, Antioxidant activity of these four peptides was evaluated by the assay methods of DPPH, ORAC, and FRAP. All these compounds have earlier been validated as antioxidant potentials and substantial data has been found online, even by authors cited in references 11-15. Then, what is the purpose of repeating all those analyses?  

4. The authors must specify the primary sources of the peptides

5. The authors must perform in vitro analysis using another neuroblastoma cell line and validate the consistent results from SH-SY5Y cells

6. The authors did not perform any in vivo experiments in the present study. They did not get consistent outcomes in the protective effect of peptides on neuronal injury animal models.

7. Minor typographical errors were found throughout the manuscript and should be amended.

Author Response

Dear Editor of International Journal Molecular Science,

Prof. Dr. Maurizio Battino

We sincerely appreciate all the suggestions and promptly improved the manuscript with all the recommended changes, answering all reviewers’ concerns, to ensure all conditions for it to be accepted for publication in International Journal Molecular Science.

We carefully addressed all reviewer’s questions and reply point-by-point, as follows:

Reviewer 1

The article by Renata Perlikowska et al. entitled “The therapeutic potential of naturally occurring peptides in counteracting SH-SY5Y cells injury” was submitted to IJMS. The reviewer's enthusiasm remains limited due to the following concerns.

  1. The authors have not followed the preparation of the manuscript according to the author’s instructions.

Answer: We thank the referee for the pertinent comment. We prepared the manuscript according to the instructions, thus the order of subsections and the numbering of the references have been changed.

  1. The author must write the aim, objectives, and conclusion of the study in the abstract

Answer: We thank the referee for the pertinent comment. The abstract was improved, we added the missing part in the abstract (Lines 15-17, 31-33).

  1. Line 35-38, Antioxidant activity of these four peptides was evaluated by the assay methods of DPPH, ORAC, and FRAP. All these compounds have earlier been validated as antioxidant potentials and substantial data has been found online, even by authors cited in references 11-15. Then, what is the purpose of repeating all those analyses?

Answer: Thank you for your observation. The authors determined the antioxidant activity for all 4 synthesized peptides. Since, rubiscolin-5 has not been tested so far, we decided to perform those assays for all compounds together under the same conditions.

  1. The authors must specify the primary sources of the peptides.

Answer: Therefore, due to the relevance of reviewer observation, we decided to improve our Results and Material and Methods section with this point of view (Lines 124 – 131 and 395-412). For many years My research group (RP, Department of Biomolecular Chemistry, Faculty of Medicine, Medical University, Lodz, Poland) is engaged in the synthesis of peptides, and the method (lines 124-131 and 395-412) described in the manuscript are implemented in our laboratory and performed on the daily basis. So far, I (RP) have synthesized endomorphin, morphiceptin, rubiscolins as the parent peptides and their linear and cyclic analogs, the results are published in the following journals: Peptides, Bioorg Med Chem., Eur J Med Chem., Prog Neuropsychopharmacol Biol Psychiatry., etc.

  1. The authors must perform in vitro analysis using another neuroblastoma cell line and validate the consistent results from SH-SY5Y cells

Answer: We thank the referee pertinent comment. We agree with the reviewer’s comment and we believe that the data presented in our study gives us a promising glimpse of neuroprotective potential of those peptides. However, it is also importance to emphasize that this study comprised a screening that allowed to identify peptides with relevant neuroprotective activities using a cellular model (SH-SY5Y cells) widely applied to elucidate the dopaminergic neurons cell death mechanisms and study the potential of new therapeutic approaches. At this point, after defining the most promising peptide, we agree with the reviewer that the compound should be tested in other cellular models, which certainly would reinforce the results observed. Therefore, further experiments should be accomplished in more relevant cellular models, such as SH-SY5Y differentiated cells and/ or using more complex models such as co-culturing systems (e.g. neuron and microglia-derived cells co-culture) or in vivo models. As a response to the reviewer’s comment, and considering the main focus of our study, we decided to address this issue in our Conclusion section (Lines 536-542).

  1. The authors did not perform any in vivo experiments in the present study. They did not get consistent outcomes in the protective effect of peptides on neuronal injury animal models.

Answer: We thank the referee for the pertinent comment. For a better understanding of the neuroprotective effects of synthesized peptides in PD model, we began to first verify their potential in the in vitro model. We agree with the reviewer that despite the promising effects displayed by peptides on in vitro cellular models, it will be required to validate these results in vivo, in order to access their pharmacological potential for neuroprotective treatment.

However, in vivo preclinical studies require important ethical and experimental considerations, including previous pharmacokinetics studies and the optimized drug formulation. Therefore, given the promising results obtained here, describing the in vitro phenotypic characterization and mechanism of action, we will proceed with the in vivo studies in an independent study, after approval by the local ethics committee and national competent authorities for animal protection. Still, we decided to improve the Conclusion section (Lines 536-542).

  1. Minor typographical errors were found throughout the manuscript and should be amended.

Answer: Thank you for your observation. All errors have been corrected.

Reviewer 2 Report

The manuscript is good and interesting but it need a section of conclusions

Author Response

Reviewer 2

The manuscript is good and interesting but it need a section of conclusions

Answer: Thank you for your comment. The changes are highlighted in the manuscript (Lines 536-542).

Reviewer 3 Report

Opioid peptides are well known for their potent neuroproptective effect. Therefore, the presented paper seems to be an important addition to existing information. However, although this manuscript is quite interesting, some issues need to be addressed before its publication

1. In the Introduction section the Authors informed that the aim of the study was to determine biological activity of naturally existing opioid peptides. In my opinion, the Authors should provide more information on these structures and their behavior towards opioidergic system. 

2. In line with this, there is no balance between information given on endomorphins and rubiscolins, as much more data is provided for endomorphins with almost lack of such in case of rubiscolins.

3. The Authors performed cytotoxic and neuroproptective studies by incubating  the cells with or without the peptides for 24 h. I would be nice if the Authors can provide additional results from the incubation for 48 or 72 h, respectively. If there is no such results, please provide the reason for which the study was not conducted after 24 h.

4. I'm quite confused with the fact that when determining the cytotoxic effect induced by the compounds, the Authors written they used these structures at non-toxic concentration. What was the reason to evaluate cytotoxicity when the Authors used concentrations known as non-toxic. 

5. Finally, the Authors show the results where each compound was evaluated in terms of its antioxidant activity only at one concentration of 10 micromol. In my opinion, the paper would be much improved if the Authors provided results for at least 3 different concentrations, and IC 50 was determined. Otherwise, there is no information why the Authors chose this exact concentration. Also, no information whether higher concentrations resulted in any lack of effect or eventual side effects is given. 

6. Fig. 2B and 2C - please introduce the results for the control. Also, please provide the statistics

7. Since the neuroproptective effect mediated by R-6 was not abolished either by mu or delta opioid receptor antagonists, what was the reason for not introducing kappa opioid receptor antagonist?

8. English should be carefully checked, as some typos and other mistakes can be found

Author Response

Reviewer 3

Opioid peptides are well known for their potent neuroprotective effect. Therefore, the presented paper seems to be an important addition to existing information. However, although this manuscript is quite interesting, some issues need to be addressed before its publication

  1. In the Introduction section the Authors informed that the aim of the study was to determine biological activity of naturally existing opioid peptides. In my opinion, the Authors should provide more information on these structures and their behavior towards opioidergic system.

Answer: We thank the referee for the pertinent comment. Therefore, due to the relevance of reviewer observation, we decided to improve our Introduction section with this point of view. The changes are highlighted in the manuscript (Lines 76-85 and 90-93).

We added an article in the References section [Ref. no. 11] and the main text [lines 85 and 93]

[11] Karasawa, Y.; Miyano, K.; Fujii, H.; Mizuguchi, T.; Kuroda, Y.; Nonaka, M.; et al. In Vitro Analyses of spinach-derived opioid peptides, rubiscolins: Receptor selectivity and intracellular activities through G protein- and β-arrestin-mediated pathways. Molecules 2021, 26, 6079.

  1. In line with this, there is no balance between information given on endomorphins and rubiscolins, as much more data is provided for endomorphins with almost lack of such in case of rubiscolins

Answer: We thank the referee for the comment. As Zadina and his colleagues [Ref. 7.Zadina, J.E., et al. Nature 1997, 386, 499–502] firstly reported the existence of endomorphins in the mammalian brain, a physiological role of endomorphins has been well established. The first reports concerning the structure and activity of rubiscolins come from 2001 [Ref. 9. Yang, S.; et al FEBS Lett. 2001, 509, 213–217], and till now their in vitro/in vivo profiles have not been sufficiently revealed. There is a lot of evidence regarding the neuroprotective effects of endomorphins [lines 97-116], and a lack of similar studies about rubiscolines, as highlighted in lines 117-119, for this reason, we decided to evaluate the neuroprotective potential of plant-derived peptides. Moreover, rubiscolins are promising in terms of their unique effects such as memory-boosting anxiolytic and antidepressant-like effects, which raises the question of whether rubiscolins, can counteract neuronal damage. Still, we decided to improve the Introduction section [lines 82-85 and 90-93]

We added an article in the References section [Ref. no. 11] and the main text [lines 85 and 93].

  1. The Authors performed cytotoxic and neuroproptective studies by incubating the cells with or without the peptides for 24 h. I would be nice if the Authors can provide additional results from the incubation for 48 or 72 h, respectively. If there is no such results, please provide the reason for which the study was not conducted after 24 h.-

Answer: Thank you for your observation regarding the time chosen to evaluate the cytotoxic activity of peptides. We agree with the reviewer that 48 h and 72 hours of treatment could be evaluated to ensure that compounds elicit their full activity. However, based on the findings reported by Larsson and collaborators (2020) [1] that carried out drug sensitivity screenings of cytotoxic drugs for 24, 48, 72 hours, the more marked effects are observed during the first 24 h. Therefore, we decided to show the cytotoxicity in the first 24 h.

[1] Larsson, P., Engqvist, H., Biermann, J. et al. Optimization of cell viability assays to improve replicability and reproducibility of cancer drug sensitivity screens. Sci Rep 10, 5798 (2020). https://doi.org/10.1038/s41598-020-62848-5

  1. I'm quite confused with the fact that when determining the cytotoxic effect induced by the compounds, the Authors written they used these structures at non-toxic concentration. What was the reason to evaluate cytotoxicity when the Authors used concentrations known as non-toxic.

Answer: Thank you for your observation. We performed cytotoxicity to be sure that selected compounds are not toxic for SH-SY5Y cells. Based on this study, we revealed the concentrations that did not exhibit cytotoxicity for SH-SY5Y having described in the manuscript as non-toxic concentrations, to search for the compound with the highest neuroprotective potential.

  1. Finally, the Authors show the results where each compound was evaluated in terms of its antioxidant activity only at one concentration of 10 micromol. In my opinion, the paper would be much improved if the Authors provided results for at least 3 different concentrations, and IC 50 was determined. Otherwise, there is no information why the Authors chose this exact concentration. Also, no information whether higher concentrations resulted in any lack of effect or eventual side effects is given.

Answer: Thank you for your observation. For FRAP and ORAC assay the concentration in the Figure 2B and 2C was a mistake because the values were extrapolated by the means of a calibration curve using as standard FeSo4 and Trolox, respectively (lines 145-146). New figures (Figure 2B and 2C) were added to the manuscript. For DPPH assay we introduced Figures 2A with statistics and in Supplementary Material Figures S1 with 3 different concentrations.

  1. Fig. 2B and 2C - please introduce the results for the control. Also, please provide the statistics.

Answer: Thank you for your observation. Please, see the previous answer. Figure 2 was corrected, additionally we added Figures S1 and Table S1 in the Supplementary section.

  1. Since the neuroproptective effect mediated by R-6 was not abolished either by mu or delta opioid receptor antagonists, what was the reason for not introducing kappa opioid receptor antagonist?

Answer: Thank you for your query. Karasawa et al. [Ref.11] showed in their functional studies that rubiscolins possessed effects only on DOR, whereas the little effect was observed on MOR and KOR. Their results showed good agreement with our opioid receptor study and functional test for receptor activation (not yet published). However, according to your suggestion, we performed experiments with kappa opioid receptor antagonist to check if the kappa receptor may play a role in neuroprotection (lines 213-232, Figure 6).

  1. English should be carefully checked, as some typos and other mistakes can be found.

Answer: We thank the referee for the pertinent comment. The typos and other mistakes were corrected.

Reviewer 4 Report

Dear authors.

Investigating the biological activity of new drugs is still an important research topic, and conducting in vitro studies often yields important new knowledge. The submitted manuscript is long and convoluted and did not sufficiently explain the main research topic

A few minor comments at the outset.

1. the manuscript presented was not prepared - formatted according to the editor's requirements, the editor's form needs to be improved.

2. When bioassays are carried out on cell culture models, it is important to take special care and ensure optimal growth conditions for the cells. For the SHSY5Y line, EMEM/F12 1:1 medium is required and not DMEM/F12. In my experience, cells do not grow properly on the medium used.

3. It is now good practice in cell culture studies to include photographic documentation showing the cell cultures to be tested and documentation of the experiments. Such documentation can be in the paper or supplement. 

Unfortunately, after reading the paper I believe that the research was not carried out correctly.

The SHSY5Y cell line was chosen correctly for this study. It is used as a model in neurobiological research. Unfortunately, these cells must be differentiated and show a phenotype characteristic of neurons. In this study, the authors unfortunately only assessed the effect of naturally occurring peptides on tumour cells. The model presented in this paper is not a neurobiological model. The work also did not use any agent that could cause toxicity. Only then can conclusions be drawn that demonstrate neuroprotection. 

The results presented in the paper can be used, but studies should be performed on differentiated cells and consideration should be given to the damaging agent - amyloid, perhaps?

Please don't break down. The paper can be improved and resubmitted for review. Unfortunately, as presented, in my opinion it should not be published. 

Author Response

Dear authors.

Investigating the biological activity of new drugs is still an important research topic, and conducting in vitro studies often yields important new knowledge. The submitted manuscript is long and convoluted and did not sufficiently explain the main research topic

A few minor comments at the outset.

  1. the manuscript presented was not prepared - formatted according to the editor's requirements, the editor's form needs to be improved.

Answer: Thank you for your comment. We prepared the manuscript according to the instructions, thus the order of subsections and the numbering of the references have been changed.

  1. When bioassays are carried out on cell culture models, it is important to take special care and ensure optimal growth conditions for the cells. For the SHSY5Y line, EMEM/F12 1:1 medium is required and not DMEM/F12. In my experience, cells do not grow properly on the medium used.

Answer: We thank the referee for the pertinent comment. The SH-SY5Y cell line used in this study was purchased from DSMZ under the code ACC 209 and the growth medium described to be used is Dulbecco's MEM. Different mediums have been used to cultivate SH-SY5Y cells. According to Xicoy and collaborators [1] the most used is DMEM followed by DMEM/F12. It is also know that the composition medium can influence the cell's response. Thus, we would like to refer that all experiments were carried out with DMEM/F12 medium avoiding any possible interference.  

[1] Xicoy, H., Wieringa, B., & Martens, G. J. (2017). The SH-SY5Y cell line in Parkinson’s disease research: a systematic review. Molecular neurodegeneration, 12(1), 1-11.

Some references that use DMEM/F12:

Lopes, F. M., Londero, G. F., de Medeiros, L. M., da Motta, L. L., Behr, G. A., de Oliveira, V. A., Ibrahim, M., Moreira, J. C., Porciúncula, L. O., da Rocha, J. B., & Klamt, F. (2012). Evaluation of the neurotoxic/neuroprotective role of organoselenides using differentiated human neuroblastoma SH-SY5Y cell line challenged with 6-hydroxydopamine. Neurotoxicity research, 22(2), 138–149.

Luo Y, Zhou S, Takeda R, Okazaki K, Sekita M, Sakamoto K. Protective Effect of Amber Extract on Human Dopaminergic Cells against 6-Hydroxydopamine-Induced Neurotoxicity. Molecules. 2022 Mar 10;27(6):1817.

Goksu Erol, A. Y., Kocanci, F. G., Demir-Dora, D., & Uysal, H. (2022). Additive cell protective and oxidative stress reducing effects of combined treatment with cromolyn sodium and masitinib on MPTP-induced toxicity in SH-SY5Y neuroblastoma cells.

  1. It is now good practice in cell culture studies to include photographic documentation showing the cell cultures to be tested and documentation of the experiments. Such documentation can be in the paper or supplement.

Answer: According with MDPI instructions available at Instructions for Authors, we state that "The data presented in this study are available on request from the corresponding author." However, we improve our manuscript with a supplementary section.

Unfortunately, after reading the paper I believe that the research was not carried out correctly.

The SHSY5Y cell line was chosen correctly for this study. It is used as a model in neurobiological research. Unfortunately, these cells must be differentiated and show a phenotype characteristic of neurons. In this study, the authors unfortunately only assessed the effect of naturally occurring peptides on tumour cells. The model presented in this paper is not a neurobiological model. The work also did not use any agent that could cause toxicity. Only then can conclusions be drawn that demonstrate neuroprotection.

The results presented in the paper can be used, but studies should be performed on differentiated cells and consideration should be given to the damaging agent - amyloid, perhaps?-

Answer: We thank the referee for the pertinent comment. We agree with the reviewer’s comment that it is of utmost importance to accomplish further studies in more relevant cellular models, such as SH-SY5Y differentiated cells and/ or using more complex models such as co-culturing systems (e.g. neuron and microglia-derived cells co-culture) or in vivo models. However, we believe that is study gives us a promising glimpse of neuroprotective potential of those peptides allowing us to outline more complex experiments to validate their therapeutic potential in the treatment of neurodegenerative diseases. As a response to the reviewer’s comment, and considering the main focus of our study, we decided to address this issue in our Discussion section (Lines 338-341) and added article wrote by Xie, H.R.; et al Chin. Med. J. 2010, 123, 1086–1092 [Ref. 25].

SH-SY5Y cell lines are widely used as a cellular model for PD research. These cells possess many characteristics of dopaminergic neurons, they express tyrosine hydroxylase and dopamine-β-hydroxylase as well as the dopamine transporter, even without being differentiated [1,2]. Generally, this cell line can be differentiated into a functionally mature neuronal phenotype in the presence of various agents (retinoic acid). Furthermore, the occurrence of several events in neuronal cells such as oxidative stress, mitochondrial dysfunction, and neuroinflammation by external or internal factors can lead to neurodegeneration of dopaminergic neurons and consequently lead to the death of neurons. In this sense, the neurotoxin 6-hydroxydopamine has been widely used by several researchers an in vitro models of Parkinson's disease, namely using SH-SY5Y cells, since it has toxic effects and leads to the occurrence of the events described above [3-5]. Therefore, in this study, this neurotoxin was used as a toxic agent and it was verified if, in the presence of the peptides, they had the capacity to reverse the effect induced by 6-OHDA.

References:

[1] Xie, H.R.; Hu, L.S.; Li, G.Y. SH-SY5Y human neuroblastoma cell line: in vitro cell model of dopaminergic neurons in Parkinson's disease. Chin. Med. J. 2010, 123, 1086–1092.

[2] Alrashidi, H., Eaton, S., & Heales, S. (2021). Biochemical characterization of proliferative and differentiated SH-SY5Y cell line as a model for Parkinson's disease. Neurochemistry International145, 105009.

[3] Elyasi, Leila, Jahanshahi, Mehrdad, Jameie, S. B., Hamid Abadi, Hatef Ghasemi, Nikmahzar, Emsehgol, Khalili, Masoumeh, Jameie, Melika and Jameie, Mana. „6-OHDA mediated neurotoxicity in SH-SY5Y cellular model of Parkinson disease suppressed by pretreatment with hesperidin through activating L-type calcium channels” Journal of Basic and Clinical Physiology and Pharmacology, vol. 32, no. 2, 2021, pp. 11-17. https://doi.org/10.1515/jbcpp-2019-0270

[4] Jaisin Y, Ratanachamnong P, Kuanpradit C, Khumpum W, Suksamrarn S. Protective effects of γ-mangostin on 6-OHDA-induced toxicity in SH-SY5Y cells. Neurosci Lett. 2018 Feb 5;665:229-235. doi: 10.1016/j.neulet.2017.11.059. Epub 2017 Nov 28. PMID: 29195909.

[5] He, Xin1,2; Yuan, Wei3; Yang, Chun-Qing4; Zhu, Lu1; Liu, Fei2; Feng, Juan2; Xue, Yi-Xue PhD1,*. Ghrelin alleviates 6-hydroxydopamine-induced neurotoxicity in SH-SY5Y cells. Neural Regeneration Research: January 2022 - Volume 17 - Issue 1 - p 170-177 doi: 10.4103/1673-5374.314314

Round 2

Reviewer 1 Report

Accept in present form

Author Response

thanks

Reviewer 2 Report

The authors added the section of conclusions

Author Response

thanks

Reviewer 3 Report

The Authors have now sufficently improved the paper. Therefore, in my opinion it is now ready to be published

Author Response

thanks

Reviewer 4 Report

Dear authors

The manuscript resubmitted for review has been formatted in accordance with the editors' requirements. The comment regarding the culture medium is dictated by my experience and the recommendations of the European Collection of Authenticated Cell Cultures. Cells do not grow properly on the medium used.

It is the reviewer's job to show errors in the work to the best of his knowledge and experience. The work must be supplemented with results on differentiated cells and supplemented with photographic documentation, which is necessary due to the culture conditions. 

Round 3

Reviewer 4 Report

Thank you for your answers. I wish you success in your further research.